# A Stricter Constraint Produces Outstanding Matching: Learning More Reliable Image Matching Using a Quadratic Hinge Triplet Loss Network[*]

Meng Xu[†]
Queen Mary University of London

Chen Shen[‡]
Didi Chuxing

Jun Zhang[§]
Didi Chuxing

Zhipeng Wang[¶]
Peking University

Zhiwei Ruan[‖]
Didi Chuxing

Stefan Poslad[**]
Queen Mary University of London

Pengfei Xu[††]
Didi Chuxing

## ABSTRACT

Image matching is widely used in many applications, such as visual-based localization and 3D reconstruction. Compared with traditional local features (e.g., SIFT) and outlier elimination methods (e.g., RANSAC), learning-based image matching methods (e.g., Hard-Net and OANet) show a promising performance under challenging environments and large-scale benchmarks. However, the existing learning-based methods suffer from noise in the training data and the existing loss function, e.g., hinge loss, does not work well in image matching networks. In this paper, we propose an end-to-end image matching method that with less training data to obtain a more accurate and robust performance. First, a novel data cleaning strategy is proposed to remove the noise in the training dataset. Second, we strengthen the matching constraints by proposing a novel quadratic hinge triplet (QHT) loss function to improve the network. Finally, we apply a stricter OANet for sample judgement to produce more outstanding matching. The proposed method shows the state-of-the-art performance when applied to the large-scale and challenging Phototourism dataset that also reported the 1st place in the CVPR 2020 Image Matching Challenges Workshop Track1 (unlimited keypoints and standard descriptors) using the reconstructed pose accuracy metric.

**Index Terms:** Image matching; HardNet; OANet; SIFT; Large scale; Challenging environments; Pose accuracy

## 1 INTRODUCTION

Image matching is the foundation task of some high-level 3D computer vision tasks, such as 3D reconstruction, structure-from-motion, camera pose estimation, etc. The goal of the task is to solve the correspondences within the same physical region from two images which have a common vision area [9]. Currently, with the widespread development of deep learning technologies applied in many computer vision areas and the need for long-term large-scale environments tasks, the shortcomings of the traditional keypoint-based image matching method become apparent. The major difficulty lies in that usually local features are evaluated through the accuracy of descriptors on small datasets, which only illustrates the matching performance, and is not suitable for integrating and evaluating some downstream tasks.

---

[*]This work is done during Meng Xu's internship at Didi Chuxing.

[†]e-mail: meng.xu@qmul.ac.uk

[‡]e-mail: jayshenchen@didiglobal.com

[§]e-mail: zhangjunmichael@didiglobal.com

[¶]e-mail: zhipengwang@pku.edu.cn

[‖]e-mail: ruanzhiwei@didiglobal.com

[**]e-mail: stefan.poslad@qmul.ac.uk

[††]e-mail: xupengfeipf@didiglobal.com

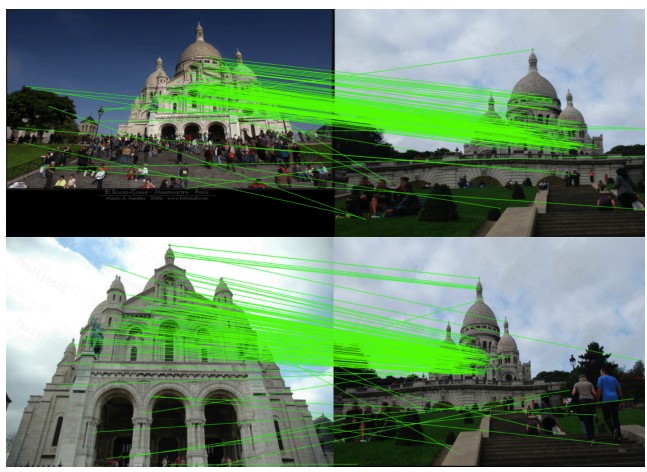

Figure 1: Our matching method's correspondences during extreme illumination variation and viewpoint changes.

The use of a deep learning-based method for image matching can overcome the disadvantages of traditional keypoint-based methods, which demonstrates the ability to integrate multi-stage tasks [11] [24]. This could be used to optimize and evaluate different evaluation metrics in large datasets. Although without using complex hand-crafted accompanied by an appropriate pipeline, the method faces challenging conditions, such as illumination variations, viewpoint changes, repeated textures that are apparent in outdoor datasets where scenes scales and conditions are highly variable. This harms performance for both accuracy and robustness.

To solve this problem, further end-to-end solutions and modified descriptors have been proposed with the premise being that the descriptor or network could learn more reliable features across image pairs. A novel representation of features with a log-polar sampling method [12] is proposed to achieve scale invariance. Some works [11] [24] propose a jointly learning feature detection and description method, which could separately improve the image matching performance.

We propose a three-stage pipeline, including feature extraction, feature matching, and outlier pre-filtering steps, to compute the corresponding pixels from image pairs, as shown in Fig. 1. For each step, we add some constraints to enforce the algorithm to converge to enable better matching. Unlike the previous methods, our proposed method only requires a lightweight model and it does not need to train on multi-branches separately.

We show that our method outperforms previous algorithms and achieves state-of-the-art performance using the Phototourism benchmark with large-scale environments and under challenging conditions. We provide a detailed insight into how an improved data processing strategy, HardNet [18] loss function, modified OANet [29] combined with guided match algorithm [2] could help the pipeline

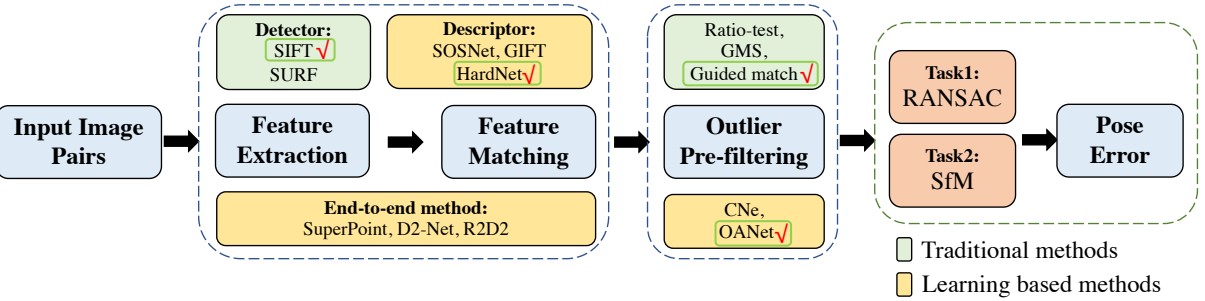

Figure 2: Image matching based pose estimation pipeline, with some popular traditional and learning-based methods. Our method is based on the pipeline with some improvement on the chosen methods (indicated by a green box with a red tick).

to achieve accurate and robust matching results and evaluate the method on a pose estimation task. Our main contributions include:

- We construct a new patch dataset based on the given Photo-tourism images and sparse model, which is similar to the UBC Phototour dataset [7], and pre-train our model on that dataset;

- We propose a novel quadratic hinge triplet (QHT) loss function for the feature descriptor network HardNet, and an improved OANet combined with a guided matching strategy to compute reliable feature matches;

- Through abundant experiments and the ablation study for each module, we show that our algorithm achieves state-of-the-art performance in the reconstructed poses, which ranks first on both tasks of stereo and multi-view matching on the public Phototourism Image Matching challenge 2020 [1].

## 2 RELATED WORK

Image matching plays an important role in many high-level tasks in computer vision area [19] [25], which is to find the corresponding pixels from the image pairs in the same physical region in which the scene imaged with geometrical constraints [9]. Feature extraction, feature matching and outlier pre-filtering are the most vital components in image matching pipeline, in which traditional methods and deep learning-based methods are implemented and completed in recent years. Fig. 2 shows the main parts of image matching based pose estimation pipeline, including some common methods and our chosen target algorithm to further improve.

Feature Extraction    End-to-end feature extraction and matching methods are classified into detect-then-describe (e.g. Super-Point [10]), detect-and-describe (e.g. R2D2 [24], D2-Net [11]), and describe-then-detect (e.g. DELF [20]) strategies according to the sequences for executing the detection and description processes, which are utilized in different needs. However, they either have poor performance on challenging conditions and lack of repeatability on keypoints or show low efficiency on matching and storage. Traditionally, detectors and descriptors are separately applied in the pipeline, SIFT [16] (and RootSIFT [3]) and SURF [4] are most popular detectors while some descriptors are followed, in which LogPolar [12] shows better performance than ContextDesc [17] from the relative pose error, while SOSNet [26] and HardNet [18] surpass GIFT [15] in the public validation set.

Outlier pre-filtering    There are a large number of incorrect matches in the matching pair, which will bring the noise to the subsequent pose estimation. Traditional outlier removal methods include ratio-test [16], GMS [5], guided-match [2] methods, etc.

A method based on deep learning is to judge whether the match is an outlier through the pose relationship obtained by the regression of the convolutional network, but it is usually difficult for network training to converge. Another way is to convert the pose to the label of whether the match is correct through the epipolar constraint, and then the regression problem can be converted into a classification problem to determine whether the match is inlier or outlier. The standard binary cross-entropy loss can be used to learn the model.

However, the input matching pairs are unordered, so the network needs to be permutation-invariant (insensitive) to match sequence transformations, so it is not applicable to convolutional networks or fully connected networks. CNe [28]draws on the idea of PointNet [23] and proposes a multi-layer weight-sharing perceptron to makes the network irrelevant to the correspondences' order transformation. For each input matching pair, the same perceptron is used to process iterations, but this also leads that each matching pair is processed independently, lacks information circulation, and cannot integrate the predicted matching pair judgment results, and thus cannot learn the useful information in the entire image matching pair. Context normalization [21] is an instance normalization method that is often used widely in image migration [14] and GAN [27]. It normalizes the processing results of image matching pairs to exchange and circulate information. But this simple and violent use of mean variance ignores the complexity of global features, and at the same time cannot capture the correlation information between parts. OANet [29] proposed the use of DiffPool and DiffUnPool layers to promote the information flow and communication of internal neurons.

To estimate a more accurate pose of the given image pairs or 3D reconstruction through the matching process, we propose improving the pipeline as follows:

- Obtain stable and accurate keypoints. The keypoints should appear invariably, e.g., a certain point in the first image could also appear stably in the second image taken under a different viewpoint or illumination change.

- Provide reliable and distinguishable descriptors. Under different environmental conditions, the same keypoint should have a similar descriptor, so that the same keypoint in different images can be paired using nearest neighbor matching.

- Hold powerful outlier pre-filtering ability. Filtering the wrong matches to get most of the correct matching pairs could help to get an accurate pose estimation result finally.

## 3 METHOD

In this paper, we aim to learn accurate and reliable image matching in large-scale challenging conditions. In contrast with previous works which use end-to-end solutions of insufficient accuracy at the feature extraction stage or traditional solvers through the image matching pipeline, we combined traditional detector solver with an improved HardNet [18] and OANet [29] to pretrain on a constructed dataset. In particular, we propose a dataset construction method to

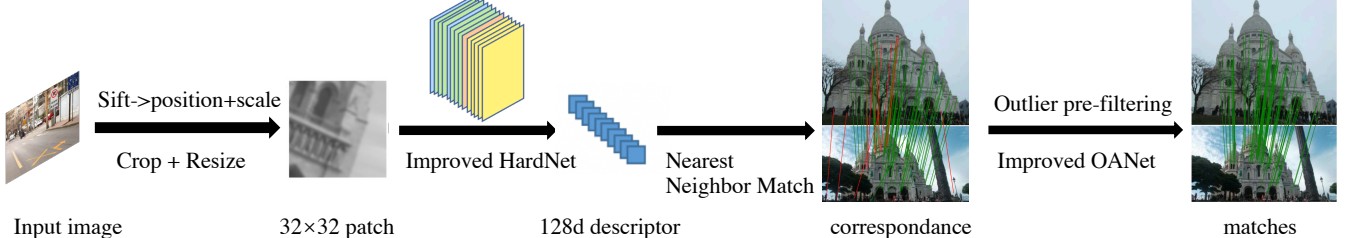

Figure 3: Architecture overview: We first use a SIFT feature detector, then we apply HardNet from a 32×32 patch input into the 128 dimension descriptors, after the nearest neighbor match and outlier pre-filtering. Then we get the final matches, which is further used to compute the pose estimation.

optimize hyperparameters before training on a large-scale dataset and propose a new loss function in the description stage based on HardNet network, during the outlier pre-filtering stage. We combined a guided match with a proposed stricter OANet to learn more accurate matches.

The whole architecture of the proposed method is presented in Fig. 3. Firstly, 128 dimensions of features are extracted from a basic HardNet with the input $32 \times 32$ patches. Then, after the matching stage, with the proposed stricter OANet, the correspondences could be transferred into an N binary classification (inlier/outlier), which is permutation-invariant and feasible with convolutional layers. Finally, the pre-filtered matches are used to compute pose estimation in stereo and multi-view tasks.

### 3.1 Dataset Construction

#### 3.1.1 UBC Dataset Construction for Pretraining

In order to quickly train our lightweight method and search for some hyperparameters to further use into the pretrained model, thus we use the UBC Phototour dataset for the pretraining, which is patch data and suitable for HardNet training.

#### 3.1.2 Phototourism Dataset Construction for Training

After pretraining on the constructed UBC Phototour dataset for the fast parameter selection during the training period, we construct a clean dataset with low noise and less redundancy from Phototourism training scenes. Removing the redundant data could largely shorten the training period significantly, while noise label reducing could enable the loss function gradient descent optimization to improve the network performance.

To reduce the noise label in the dataset, we set the threshold to discard the images with low confidence, in which with the least 25% 3d points are tracked less than 5 times. When removing redundant data, we sample the 3d points with tracks results that have been tracked more than 5 times. The sampling is repeated 10 times, and the Normalized Cross-correction (NCC) value is calculated for each sample (the lower of the NCC indicates the lower similarity of the two images and the larger the difference of the matching), and the result with the lowest NCC is retained. In addition, we also applied data augmentation with random flip and random rotation strategies in both pretraining and training processes.

### 3.2 Feature Extraction with Improved HardNet

Feature extraction includes keypoint detection and feature description processes. SIFT detector [16] is used firstly to extract the position and scale information for the selected keypoint of the given input image. We adopted the OpenCV implemented SIFT with a low detection threshold to generate up to 8000 points and a fixed orientation[1]. A single pixel on the image cannot describe the physical information of the current key point, so in order to describe the key

---

[1] https://github.com/vcg-uvic/image-matching-benchmark-baselines

point and the information around it, the key point is cropped with its scale size, called a patch, and then resized to $32 \times 32$ as the input to the HardNet network to extract the patch's descriptor.

After a relatively simple 7-layer HardNet, the input 32×32 patch is described as a 128-dimensional feature vector, which represents the description generated for each patch. We retain the network structure and made improvements to its loss function to train the network stably and efficiently on the constructed dataset.

HardNet uses a triplet loss embedded with difficult sample mining, to optimize it so that the distance of the sample descriptor within the class is small, and the distance in the sample description vector between the classes is large.

In order to further improve the effectiveness and stability of model learning, we apply a quadratic hinge triplet (QHT) loss based on the triplet loss, which is inspired by the first order similarity loss in SOSNet [26]. This is done by adding the square term of the triple, which shares the same mining strategy as used in HardNet to find the "hardest-within-batch" negatives. The description loss function $\mathscr{L}_{des}$ is expressed as:

$$\mathscr{L}_{des} = \frac{1}{N} \sum_{i=1}^{N} max(0, 1 + d_i^{pos} - d_i^{neg})^2 \tag{1}$$

$$d_i^{pos} = d(a_i, pos_i) \tag{2}$$

$$d_i^{neg} = \min_{\forall j \neq i}(d(a_i, neg_j)) \tag{3}$$

Where $d()$ stands for the L2 distance, $d_i^{pos}$ represents the distance between anchor descriptor and positive descriptor, $d_i^{neg}$ represents the minimum distance in all the distances between the anchor descriptor and the negative descriptor. The positive sample represents the patch on different images corresponding to the same 3d point with the given anchor in the real world. On the contrary, the negative sample represents the patch obtained from different 3d points.

Quadratic triplet loss weights the gradients with respect to the parameters of the network by the magnitude of the loss. Compared with HardNet, the gradient of QHT loss is more sensitive to changes in loss. A larger $d_i^{pos} - d_i^{neg}$ results in a smaller loss gradient, thus the model learning is more effective and stable.

In addition, the improved model could be sensitive to the noise in the data. Incorrectly set positive and negative sample labels will degrade the performance of the model, this risk could be avoided through the previously proposed data denoising technique.

### 3.3 Stricter OANet with Dynamic Guided Matching

OANet proposed to use Diffpool and DiffUnPool layers on top of the CNe network [28] to promote the information circulation and communication of the internal neurons. The two networks can be abstracted as seen in Fig. 4. N matching pairs with D-dimensional features are processed by the perceptron of the CNe network to obtain the output results for the same dimension. The OANet network

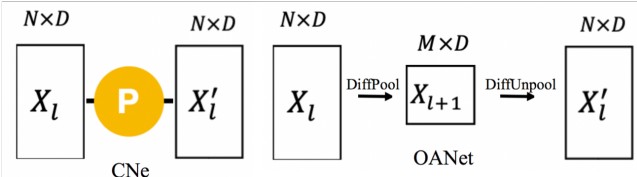

Figure 4: Our matching method's correspondences in extreme illumination variation and viewpoint changes.

reduces the input dimension to an M×D matrix through the Diffpool layer, and then raises the dimension back to an N×D output through DiffUnpool, and merges the information in the middle. Diffpool maps the input N matches to M by learning the weights of a soft distribution to aggregate the information, and then DiffUnpool reorganizes the information back to N dimensions. The network is not sensitive to disordered disturbances of input matching pairs.

We convert the accuracy of the pose to the accuracy of the matching to learn, so the accuracy of the matching will affect the learning ability of the model. We have made the following improvements to OANet to make it stricter to learn from positive samples and to make matching judgments the label be more accurate, which effectively filters the exception points and improves the matching and pose estimation performance. We shortened the threshold condition of the geometric error constraint from 1e-4 to 1e-6. In addition, we also added a point-to-point constraint based on the OANet point-to-line constraint. A point is judged as a negative sample when the projection distance is greater than 10 pixels.

Generally, inadequate matches may lead to inaccurate pose estimation. Apart from OANet, for the image pairs with less than 100 matches, we also proposed dynamic guided matching (DGM) to increase matches. In contrast to traditional guided matching [2], the dynamic threshold is applied using the Sampson distance constraint according to the number of matches of an image pair. We argue that a smaller number of matches requires a greater threshold. The dynamic threshold th is set by the formula as:

$$th = th_{init} - \frac{n}{15} \qquad (4)$$

where $th_{init}$ is 6 and n is the number of original matches of an image pair. For the image pairs with more than 100 matches, we directly applied DEGENSAC [4] to obtain the inliers for submission.

### 3.4 Implementation Details

We train the proposed model integrated with improved SIFT, HardNet, OANet on the Phototourism dataset [1], in which HardNet is pretrained on the constructed UBC Patch dataset [7]. The whole model is implemented with Pytorch [22] on a NVIDIA Titan X GPU. All models were trained from scratch and no pre-trained models were used. For the stereo task, the co-visibility threshold is restricted to 0.1 while for the multi-view task, the minimum 3D points are to set to 100, while no more than 25 images are evaluated at a time.

In the description stage, the input patch size is cropped to $32 \times 32$ with the random flip and random rotation. Optimization was done by an SGD solver [6] with a learning rate of 10 and weight decay of 0.0001. The learning rate was linearly decayed to zero within 15 epochs.

In the outlier pre-filtering stage, for training, we employed a dynamic learning rate schedule where the learning rate was increased from zero to the maximum linearly during the warm-up period (the first 10,000 steps) and decayed step-by-step after that. Typically, the maximum learning rate was 1e-5, and the decay rate was 1e-8. Besides, we trained the fundamental estimation model by setting the value of parameter *geo_loss_margin* to 0.1. The side information

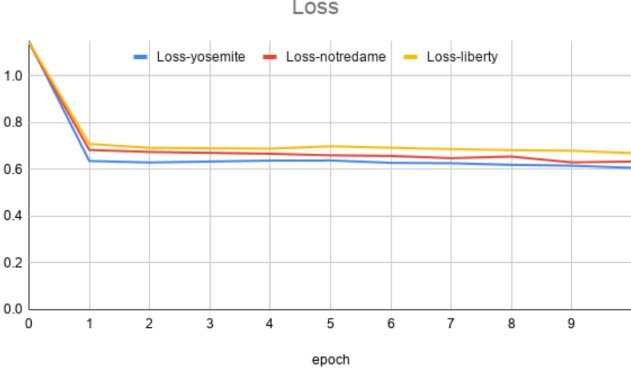

Figure 5: The loss for the training model in different scenes of the UBC Phototour dataset. The difference between different losses is not big, indicating that the improved loss is stable for the model training.

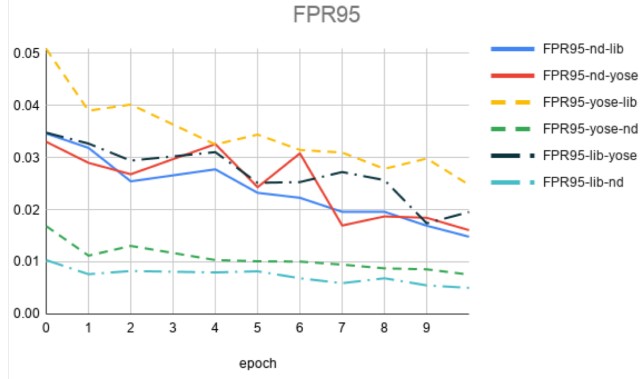

Figure 6: FRP95 metric trend, using different scene combinations of the UBC phototour dataset as the training set and the validation set. The results show that for the same validation set, the FRP95 trained on the Yosemite scene is higher, indicating that there are false matches or difficult matches.

of ratio test [16] with a threshold of 0.8 and mutual check was also added into the model input. During inference time, the output of the model will undergo a DEGENSAC to further eliminate unreliable matches. The use of DEGENSAC in a dynamic guided match shares the same configuration with the iteration number of 100000, error type of Sampson, inlier threshold of 0.5, confidence of 0.999999 and enabled degeneracy check.

## 4 EXPERIMENTS

In this section, we evaluate our proposed method on the public Phototourism benchmark with the features results. The final pose estimation results are computed online. We firstly conduct performance for several separate modules, then qualitatively and quantitatively analysis the effectiveness of our method.

### 4.1 Experimental Settings

#### 4.1.1 Datasets

**UBC Phototour dataset** [7] is a dataset which involves corresponding patches from the 3D reconstruction of Liberty, Notre Dame and Yosemite. Every tour consists of several bitmap images at the resolution of $1024 \times 1024$ pixels, each image contains image patches as a $16 \times 16$ array, in which the patch is sampled as a $64 \times 64$ grayscale. In addition, the matching information (sample 3D point index of each patch) and keypoint information (reference image

index, orientation, scale, and position) are supplied separately in the file.

**Phototourism dataset** [1] was collected from multi-sensors in 26 popular tourism landmarks. The 3D reconstruction ground truth is obtained with Structure from Motion (SfM) using Colmap, which includes poses, co-visibility estimates and depth maps. The dataset is divided into a training set (16 scenes), a validation set (3 scenes of the training set) and a test set (10 scenes). As a benchmark large-scale dataset, the scenes contained in the dataset includes various challenging conditions, e.g., occlusions, changing viewpoints, different time, repeated textures, etc.

### 4.1.2 Tasks and Evaluation metrics

We implement and compare our method with different approaches with respect to two downstream tasks: stereo and multi-view reconstruction with SfM, they both evaluate the intermediate metrics for further comparisons. The two tasks evaluate different formats of the dataset, which is subsampled into smaller random subsets from the Phototourism dataset. Stereo task evaluates image pairs while the Multi-view task evaluates 5 25 images with a SfM reconstruction. Stereo task uses the random sampling consensus method (RANSAC) [13] to estimate the matches between the two images that meet the motion consistency, thereby decomposing the pose rotation R and translation t. Multi-view task recovers the pose R and t of each image through a 3D reconstruction. By calculating the cosine distance of the vector between the estimated pose and the ground-truth pose, the performance of image matching is measured by the size of the computed angle.

Given an image pair with co-visibility constraints in the stereo task or the 3D reconstruction from multi-view images, we may compute the following metrics in different experiments. The final results are compared with the mAA metric.

- Mean average accuracy (%mAA) is computed by integrating the area under curve up to a maximum threshold, which indicates the angular angle between the ground-truth and the estimated pose vector.

- Keypoint repeatability (%Rep.) measures the ratio of possible matches and the minimum number of key points in the co-visible view.

- Descriptor matching score(%MS) is the average ratio of correct matches and the minimum number of key points in the co-visible view.

- Mean absolute trajectory error (%mATE) measures the average deviation from the ground truth trajectory per frame.

- False positive rate (%FPR) is the ratio between the number of negative events wrongly categorized as positive and the total number of actual negative events

### 4.2 Qualitative and Quantitative Results

The data construction information for the Phototourism dataset is given in Table 1, it could be seen that the proposed dataset construction method could largely reduce the amount of dataset size thus further speed up the training process and help improve the model performance. The training loss and FRP95 metric trend are shown in Fig. 5 and Fig. 6, using the UBC Phototour dataset, which indicates the loss stability and helps to understand the data characteristics. The comparison of matching visualization performance for the stereo task is shown in Fig. 7, in which it is seen that our proposed method outperforms the traditional RootSIFT based method, and with more accurate matches. Fig. 8 shows the visualization on Phototourism dataset on stereo task and multi-view task.

To validate the performance of loss of different scenes, Table 2 lists the evaluation of correspondence on UBC Phototour dataset,

Table 1: Dataset construction information comparison of two selected scenes and the average of Phototourism dataset

| Scenes | Type | Images | 3D Points | Patches |
|---|---|---|---|---|
| Buckingham_palace | original | 1676 | 246035 | 4352977 |
| | sampled | 1257 | 72814 | 364070 |
| Grand_place_Brussels | original | 1080 | 209550 | 3206171 |
| | sampled | 810 | 78108 | 390540 |
| All scenes(Average) | original | 1183.9 | 165124.8 | 3567695 |
| | sampled | 887.6 | 63433.6 | 317168 |

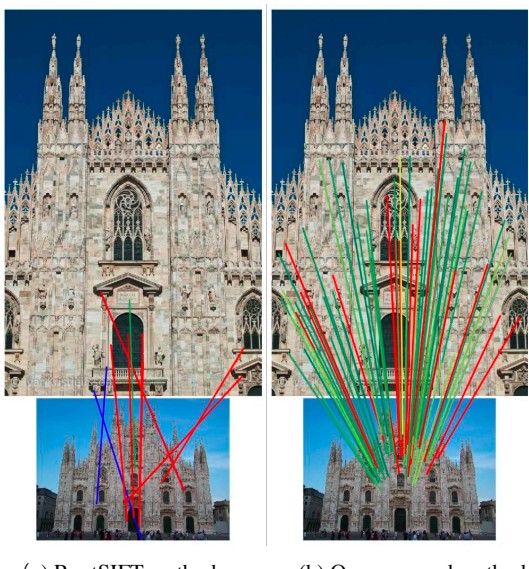

(a) RootSIFT method  (b) Our proposed method

Figure 7: Visualization of the correspondence for the stereo task based on a traditional RootSIFT based method and our proposed method (green represents correct matches, yellow encodes a match error within 5 pixels, red shows the wrong matches).

which indicates the effects and improvements of our proposed QHT loss compared to HT (hinge triplet) loss. Table 3 and Table 4 demonstrate our submitted final results on Phototourism challenge Track1, we rank 1st on both tasks.

### 4.3 Ablation Study

To fully understand each module in our proposed method, we evaluate different modules, above, using the Phototourism validation dataset. In Table 5, our method is compared with several versions to see how data construction and an improved HardNet could help to improve the descriptor and to further enhance the performance for both the multi-view and stereo tasks. We evaluate the following four variants as the feature descriptor while the other parts remain the same: 1) RootSIFT; 2) the original pretrained HardNet; 3) our proposed improved HardNet; 4) our proposed improved HardNet with a data construction technology. The comparison of different feature descriptors indicates that the model performance increases largely by adding the loss constraints on HardNet. It shows an 8% improvement in the average mAA of both tasks. With data construction, the improved HardNet yields the best performance, obtaining an average mAA of 0.7894.

To evaluate whether or not the proposed modified OANet could help improve the performance for outlier pre-filtering, Table 6 lists the performance comparison of four different outlier pre-filtering

Table 2: Patch correspondence evaluation performance with the metric of FPR95. We compare different loss functions on UBC Phototour dataset with HardNet as the description network (* means the network is trained by our implement). The best results are in **bold**.

| Train | Liberty | | NotreDame | |
|---|---|---|---|---|
| Test | Notre-Dame | Youse-mite | Liberty | Youse-mite |
| HardNet [18] | 0.62 | 2.14 | 1.47 | 2.67 |
| HardNet-HT [18] | 0.53 | 1.96 | 1.49 | 1.84 |
| HardNet-HT* | 0.5 | 1.96 | 1.48 | 1.61 |
| HardNet-QHT* | **0.45** | **1.83** | **1.228** | **1.52** |

Table 3: Phototourism challenge for the Stereo task result. Mean average accuracy for pose estimation with an error threshold of 10°. The results are submitted with an online evaluation (We only retain the result with the highest rank in the submitted results from each team).

| Method | $\%Rep.^{@3pix}$ | $MS^{@3pix}$ | $mAA^{@10°}$ | rank† |
|---|---|---|---|---|
| Ours | 0.486 | 0.823 | 0.61081 | 1 |
| Cavalli et al. [8] | 0.442 | 0.828 | 0.58300 | 12 |
| Jiahui et al. [1] | 0.487 | 0.846 | 0.57826 | 17 |
| Ximin et al. [1] | 0.486 | 0.871 | 0.58870 | 5 |

Table 4: Phototourism challenge for the Multi-view task result. Mean average accuracy in pose estimation with an error threshold of 10°. The results are submitted with an online evaluation. (We only retain the result with the highest rank for the submitted results from each team)

| Method | mATE | $mAA^{@10°}$ | rank† |
|---|---|---|---|
| Ours | 0.358 | 0.78288 | 2 |
| Cavalli et al. [8] | 0.361 | 0.77056 | 3 |
| Jiahui et al. [1] | 0.367 | 0.77041 | 5 |
| Ximin et al. [1] | 0.386 | 0.75127 | 14 |

Table 5: Ablation study of proposed HardNet with the data construction method applied to the Phototourism validation set.

| Method | $mAA^{@10°}$ | | |
|---|---|---|---|
| | Stereo | Multi-view | Avg. |
| RootSIFT | 0.6698 | 0.7258 | 0.6978 |
| Pretrained HardNet | 0.7317 | 0.7924 | 0.7621 |
| Improved HardNet | 0.7285 | 0.8159 | 0.7722 |
| Improved HardNet+CleanData | 0.7537 | 0.8252 | 0.7894 |

Table 6: Ablation study of proposed OANet method on Phototourism validation set.

| Method | $mAA^{@10°}$ | | |
|---|---|---|---|
| | Stereo | Multi-view | Avg. |
| RootSIFT+RT+CC | 0.6698 | 0.7258 | 0.6978 |
| Pretrained HardNet+RT+CC | 0.7317 | 0.7924 | 0.7621 |
| Improved HardNet+RT+CC | 0.7537 | 0.8252 | 0.7894 |
| Improved HardNet+OANet | 0.7918 | 0.8658 | 0.8288 |

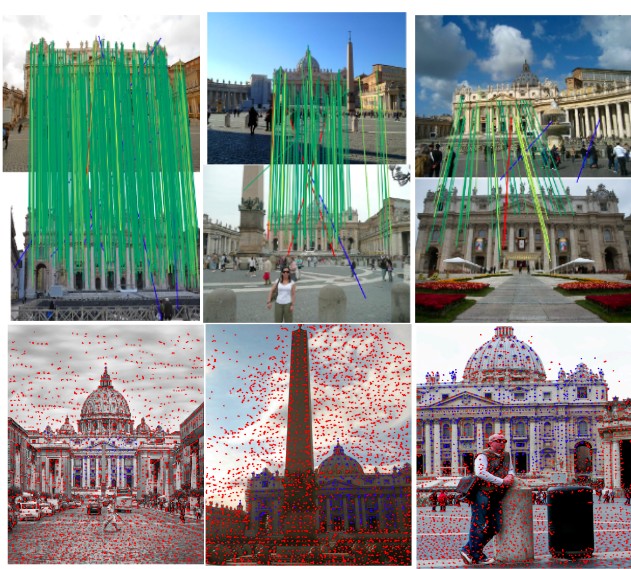

Figure 8: Visualization of correspondence for the stereo task and reconstruction points in the multi-view task. (green represents a correct match, yellow encodes a match error within 5 pixels, red shows the wrong match, blue shows the keypoints with the correct 3D points).

methods: 1) using RootSIFT as the feature descriptor while ratio test and cross check are used for the outlier filtering; 2) use of the pretrained HardNet with a ratio test and cross check; 3) use of the proposed improved HardNet with ratio test and cross check; 4) use of proposed improved HardNet with a modified OANet. The comparison of different outlier removals indicates that the performance of modified OANet surpasses the ratio test with the cross check method by a 4% in average mAA.

## 5 CONCLUSION

This paper presents a novel image matching approach, which: integrates a data construction method to remove noise and to ensure more efficient training; uses an improved HardNet QHT loss function to make the network more sensitive to noise gradient descent; uses a stricter OANet combined with guided match algorithm to reduce the recall rate of outlier pre-filtering. By proposing the QHT loss, which strengthens the distance constraints for the improved OANet with a stricter judgment on positive samples, an improved outstanding matching is produced using these stricter constraints. The proposed method is crucial to obtain high-quality correspondences when applied to a large-scale challenging dataset with illumination vibrances, viewpoint changes and repeated textures. Our experiments show that this method achieves a state-of-the-art performance on the public Phototourism Image Matching challenge.

### ACKNOWLEDGMENTS

This research was funded in part by a Ph.D. scholarship funded jointly by the China Scholarship Council and QMUL. We thank our colleagues from Didi Chuxing who provided insight and expertise that greatly assisted the research, especially for Zhongkun Chen, Wei Shao, Jingchao Zhou, Tengfei Xing and Bin Xu.

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
