# OpenReview forum: "A Stricter Constraint Produces Outstanding Matching: Learning Reliable Image Matching with a Quadratic Hinge Triplet Loss Network"
_graphicsinterface.org/Graphics_Interface/2021/Conference/Second_Cycle — GI 2021_

### Official Review · Reviewer_F8zx · 2021-05-03
**Effective improvements to image matching with impressive accuracy boost.**

**Rating:** 8
**Confidence:** 2

**Review:**

The paper describes several improvements on image matching with respect to key point selection, descriptors and filtering and loss function. They can show that their enhanced method is able to outperform state of the art approaches. Unfortunately I do not know the current literature on this specific topic well enough to judge novelty but their design decisions seem to be well motivated and explained, albeit at a quite technical level that assumes familiarity with the basic concepts.
I propose to accept the submission due to its demonstrated benefits over the state of the art.

---

### Official Review · Reviewer_Jux6 · 2021-05-04
**Interesting paper, well explained, state-of-the-art results**

**Rating:** 8
**Confidence:** 2

**Review:**

The paper presents a novel learning approach for image matching. The novelty comes from several improvements to key components of the image matching method. The paper is nicely written and easy to follow. It has a solid overview of the related work, a strong motivation, and a detailed ablation study. The only thing missing is some discussion on the limitations of the presented method. The work is tested on UBC Phototour and Phototourism datasets. How would it perform on other kinds of datasets? Are there particular features where the method fails? Can the method be further improved?

Minor comment: In the abstract, in the first sentence, is “3D construction” a typo? Should it say “3D reconstruction”?

---

### Official Review · Reviewer_8Yky · 2021-05-06
**Strong results but very poor writing**

**Rating:** 5
**Confidence:** 3

**Review:**

### Summary

The paper proposes careful enhancements to an imaging matching pipeline consisting of:
1) applying noise removal to get a cleaner patch dataset by taking the Phototourism dataset and removing images with low confidence and sampling points that has been track more than 5 times
2) improvements to the network: improved feature extraction using HardNet by using a quadratic hinge triplet (QHT), and improved constraints and matching to OANet
3) Set of experiments and ablation study on the Phototourism Image Matching challenge at CVPR 2020


### Strengths
- The proposed pipeline is able to achieve strong performance (1st place in the CVPR 2020 Image Matching Challenges Workshop Track 1)
- The strong performance results suggests that the improvements proposed by the work (while minor), was carefully considered and would be of potential interest and value to the community

### Weaknesses

- The main weakness of the work is the extremely poor writing quality.  There is spelling, grammar, and wording issues throughout (see Comments for some examples).  The writing is sufficiently poor that it makes the specifics of the proposed method difficult to understand, and aspects of the experiments hard to interpret.

- Due to the poor writing, some details are also not made clear.  For instance, the Phototourism benchmark has leaderboards for variations of unlimited/restricted keyboards and different sized descriptors.  It is not specified that the work is number one on the leaderboard for the unlimited keypoints and standard descriptors.  In addition, the work fails to discuss and reference the other works that appears in Table 3 (Luca et al, Jiahui et al, Ximin et al are not included in the reference).  It's unclear what works these refers to (Luca et al, should be Cavalli et al, AdaLAM: Revisiting Handcrafted Outlier Detection, 2020)

- It's not clear what the FRP95 metric is that is show in Table 2, Figure 6.  Note that several metrics described in section 4.1 (%Rep., %M.S., %mATE, %FPT) does not appear to be reported on in the paper.


### Comments

Some examples of misspellings, grammar and other wording issues.  A full editing pass is recommended.

Abstract
- "Traditional local features" => "traditional local features"
- "suffer the noise" => "suffer from noise"
- "less training dataset" => "less training data"

Section 1: Introduction
- "Phtotour" => "Phototour"

Section 2: Related work
- "overpass" => "surpass"
- "pointNet" => "PointNet"
- "irrelevant transformation problem" =>  ??
- "we consider improving the pipeline with respect to the following aspects" => "we propose improving the pipeline as follows"

Section 3: Method
- Figure3: "Ourlier" => "Outlier"
- "hypermeters" => "hyperparameters"
- Section 3.1.2 is poorly written. I can't really understand what is meant by: "in which 25% with the least 3d points" or "we sample the 3d points that have been tracked more than 5 times, and only keep the 3d points which are tracked 5 times".  Also, what is NCC?

- Section 3, 3.2, 3.3 should include citations to the prior work (HardNet[8], [OANet[9], and CNe [12])
- "on the basis of CNe" => "on top of the CNE network [12]"
- "with D-dimensional" => "with D-dimensional features"
- "rigorous to learn" => unclear what this means

Section 4 (Experiments)
- Table 2: "HartNet" => "HardNet"
- Section 4.3: "pret rained" => "pretrained"

References should be proofread
- Not all authors are included (et, al. used after 3rd author)
- Poorly formatted, with [J]. or [C]// before publication venues

---

### Meta-Review · Area_Chair_TXkT · 2021-05-06

**Recommendation:** Accept
**Confidence:** 3

**Metareview:**

The paper presents a novel method for image matching. The core strength of this paper is that the method exhibits improved performance over the other state-of-the-art algorithms applied on the Phototour dataset. Considering all the reviewer’s comments, this strength seems to qualify this paper for acceptance, as it may be of interest and value to the community. However, major revision is needed on the grammar, spelling, and wording in the paper writing, as well as an explanation of metrics used to calculate the performance. Furthermore, it is necessary to include comments on the limitations of the method and some deeper discussion on the performance and other methods involved in the comparison.

---

### Decision · Program_Chairs · 2021-05-08

Accept